# Attitudes towards Receiving Monkeypox Vaccination: A Systematic Review and Meta-Analysis

**DOI:** 10.3390/vaccines11121840

**Published:** 2023-12-12

**Authors:** Mostafa Hossam-Eldin Moawad, Amira Mohamed Taha, Dang Nguyen, Mohammed Ali, Yasmine Adel Mohammed, Wesam Abd El-Tawab Moawad, Esraa Hamouda, D. Katterine Bonilla-Aldana, Alfonso J. Rodriguez-Morales

**Affiliations:** 1Clinical Department, Faculty of Pharmacy, Alexandria University, Alexandria 21544, Egypt; mh3912214@gmail.com; 2Faculty of Medicine, Suez Canal University, Ismailia 41522, Egypt; 3Faculty of Medicine, Fayoum University, Fayoum 63514, Egypt; am7529@fayoum.edu.eg; 4Massachusetts General Hospital, Corrigan Minehan Heart Center, Harvard Medical School, Boston, MA 02215, USA; dnguyen61@mgh.harvard.edu; 5Faculty of Medicine, Al-Azhar University, Cairo 11884, Egypt; mohammedali19983@gmail.com; 6Faculty of Medicine, Assiut University, Assiut 71524, Egypt; yasminea126@gmail.com; 7Faculty of Pharmacy (Girls), Al-Azhar University, Cairo 11884, Egypt; wesam@mars-global.co.uk; 8MARS Global, London WC2H 9JQ, UK; 9Faculty of Medicine, Menoufia University, Menoufia 32511, Egypt; esraa.hamouda14173@med.menofia.edu.eg; 10Research Unit, Universidad Continental, Huancayo 15311, Peru; 11Clinical Epidemiology and Biostatistics Master Program, Faculty of Health Sciences, Universidad Científica del Sur, Lima 15097, Peru; arodriguezmo@cientifica.edu.pe; 12Gilbert and Rose-Marie Chagoury School of Medicine, Lebanese American University, Beirut P.O. Box 36, Lebanon

**Keywords:** monkeypox, Mpox, vaccine, willingness, hesitancy, acceptance, meta-analysis

## Abstract

Background: The public’s attitude towards Mpox vaccination is a critical factor in the success of immunisation programmes. Understanding the factors contributing to vaccine acceptance or hesitancy is critical for developing effective health communication strategies. This systematic review and meta-analysis aims to bring together evidence from observational studies on attitudes towards Mpox vaccination, including willingness and rejection. Methods: From this review’s inception until June 2023, a comprehensive search was conducted across four major electronic databases: PubMed, Web of Science, Scopus, and EBSCO. The inclusion criteria included studies investigating public attitudes towards Mpox vaccination, as defined by acceptance and willingness to be vaccinated versus rejection and unwillingness. Results: Thirty studies met the inclusion criteria among the screened literature. An analysis of 27 studies involving 81,792 participants revealed that 45,926 (56.14%) were willing to receive the Mpox vaccination. In contrast, ten studies involving 7448 participants revealed that 2156 people (28.94%) were unwilling to receive the Mpox vaccination. Females were less willing to receive the vaccine than males, with an odds ratio (OR) of 0.61 (95% CI, 0.43–0.86). Furthermore, homosexuals were found to be more willing than heterosexuals, with an OR of 1.44 (95% CI, 1.14–1.80). Conclusion: Vaccination is emerging as a critical strategy for preventing Mpox infection and fostering herd immunity against potential outbreaks. Improving public awareness and acceptance of vaccination is critical to avoiding a situation similar to the COVID-19 pandemic. Targeted educational and outreach programmes could explain the benefits of vaccination, bridging the information gap and encouraging a proactive public health approach to emerging infectious diseases.

## 1. Introduction

Mpox (monkeypox) is a re-emerging zoonotic disease caused by the human Mpox virus (MPXV), which is a complex DNA virus of the genus *Orthopoxvirus* [1]. It was identified for the first time in 1958 among monkeys in Denmark [2]. The human infection began in 1971 with a case in the Democratic Republic of the Congo, after which the disease spread to Central and West Africa. Notably, the disease spread beyond the African continent for the first time in 2003 [3,4].

Humans, although accidental hosts, are exposed to the infection directly or indirectly. Contact with infectious animal fluids, bites, or the consumption of uncooked animal flesh are all examples of direct transmission [5]. On the other hand, human-to-human transmission occurs through close or intimate contact with infectious skin lesions, fluids, respiratory droplets, or sexual contact, the latter of which is recognised as a significant risk factor. According to CDC data, many Mpox cases are linked to homosexual and bisexual men. Nonetheless, infection susceptibility extends to any individual who comes into contact with infected entities, regardless of sexual orientation [6].

A critical transmission route is maternal-to-foetal, in which the Mpox virus crosses the placental barrier and causes congenital infections. Contact with contaminated clothing, linens, or household items can also result in indirect transmission [2]. Manifestations of Mpox include a maculopapular rash, fever, headache, fatigue, and muscular discomfort [2]. Despite a symptomatic resemblance to smallpox, Mpox is distinguished by lymphadenitis, most commonly in the submandibular, submental, and inguinal regions [7].

Mpox infection cases increased across 30 countries in May 2022 [7]. The World Health Organisation (WHO) declared a multinational outbreak by the end of July, raising global alarm. As of August 10, 2023, the global case count had surpassed 89,308, with 152 fatalities recorded [8,9]. Recent increases in Mpox infection cases in various countries have highlighted the need for effective public health interventions to reduce the disease’s spread and impact.

Vaccination remains a critical strategy for preventing the spread of infectious diseases, effectively establishing herd immunity and preventing outbreaks. There is currently no Mpox vaccine available in the prophylactic landscape. However, according to the WHO and CDC recommendations, two existing smallpox vaccines, JYNNEOS and ACAM2000, have an 85% efficacy against Mpox [9]. Knowledge, cultural beliefs, previous vaccination experiences, and perceived hazards and benefits could all influence people’s attitudes and intentions towards receiving Mpox vaccination. Furthermore, given the recorded gaps in health-seeking behaviours and vaccine acceptance among different demographic groups, sexual and gender identities could have a role in influencing these attitudes.

Understanding public attitudes towards Mpox vaccination is critical for developing targeted educational and vaccination campaigns in the context of a strained global healthcare system following the COVID-19 pandemic. This will help increase vaccine uptake, slow disease spread, and decrease the risk of further strain on medical resources.

The main objective of this systematic review and meta-analysis is to compile existing literature studies to provide a comprehensive understanding of public attitudes towards Mpox vaccination. It will also investigate the possible impact of sexual and gender identities on these attitudes, providing information that could be useful in adapting initiatives to improve vaccine acceptance and coverage against this resurgent infectious threat.

## 2. Methods

This systematic review and meta-analysis was performed according to the Preferred Reporting Items for Systematic Reviews and Meta-Analyses (PRISMA) guidelines [10]. We registered the study protocol in PROSPERO (CRD42023451945).

### 2.1. Search Strategy

We searched four electronic databases (PubMed, Web of Science, Scopus, and EBSCO) from inception until June 2023. The following search strategy was used for all the databases: “(monkeypox OR Mpox OR mpxv) AND (vaccination) OR (vaccinated) OR (vaccine)”. Our search strategy was comprehensive, with no age, setting, or publication date restrictions.

### 2.2. Inclusion and Exclusion Criteria

We included studies investigating the attitudes of the people towards the Mpox vaccination, whether there is acceptance and willingness or a rejection and unwillingness to be vaccinated. We included all types of observational studies (cohort, case–control, and cross-sectional), while we excluded narrative and systematic reviews, meta-analyses, case reports, and case series.

### 2.3. Study Selection and Data Extraction

Four authors independently conducted the screening process in two steps: title and abstract screening and full-text screening to determine the final included studies. Any disagreements between the authors during the screening process were resolved by discussion with a senior author. Data were extracted in a preformed Microsoft Excel spreadsheet, which included study information (study design, year, country, sample size, and summary of findings), participant characteristics (gender, age, sexual orientation, population, preexisting diseases, and taking COVID-19 vaccination), and lastly, patients’ acceptance or refusal to receive Mpox vaccine.

### 2.4. Quality Assessment

We used the Newcastle–Ottawa scale to assess the included studies’ quality [11]. Four researchers conducted this assessment, and any disagreement was resolved with a senior reviewer. Scores of 0–3 were considered low quality, 4–6 as moderate quality, and 7–9 as high quality.

### 2.5. Statistical Analysis

We conducted the statistical analysis using Open Meta Analyst software to calculate the willingness and non-willingness rates to receive Mpox vaccines among all the participants in each study. We also used Review Manager version 5.4 to examine the factors affecting willingness to take vaccines. We used the pooled odds ratio analysis to determine the factors with higher odds of willingness to receive the vaccine. This analysis was conducted at a confidence level of 95% and a *p*-value of 0.05. Heterogeneity was assessed using I^2^ and a *p*-value of 0.05. We conducted subgroup analysis between different included populations using Open Meta Analyst software version 5.26.14.

## 3. Results

### 3.1. Literature Search Results

Our literature database search yielded 1580 records. After removing duplicates, 1202 studies remained for the title and abstract screening. Forty-nine articles were eligible for full-text screening. From these, 30 studies [1,12,13,14,15,16,17,18,19,20,21,22,23,24,25,26,27,28,29,30,31,32,33,34,35,36,37,38,39,40] were included in the meta-analysis. The PRISMA flow diagram of the study is shown in Figure 1.

### 3.2. Baseline Characteristics

All the included studies were of cross-sectional design and were conducted in different regions such as Asia, Europe, Middle East, Africa, and North America. The population differed across the studies where some were conducted on the general population, other studies on HCWs, university students, and LGBTQ (lesbian, gay, bisexual, transgender, queer and questioning) community (Table 1).

### 3.3. Quality Assessment

According to NOS, six [13,18,24,25,27,37] of the included studies were of high quality, twenty-one [1,12,14,15,16,17,19,20,21,22,26,28,29,30,31,32,34,36,38,39,40] of moderate quality, and three [23,33,35] of low quality (Table 2).

### 3.4. Meta-Analysis

#### Willingness to Receive Mpox Vaccines

Among 30 of the included studies, 27 studies investigated the willingness to receive Mpox vaccine with a total of 81,792, of which, 45,926 (56.14%) of them were willing to receive Mpox vaccination, as shown in Figure 2, while only 10 of the included studies reported vaccine refusal with a total of 7448 participants, and 2156 (28.94%) were not willing to receive Mpox vaccination, as shown in Figure 3.

### 3.5. Gender Difference in Willingness to Be Vaccinated against Mpox

Six studies reported the different genders’ willingness to Mpox vaccine. The pooled OR indicated that females are less willing to receive the vaccine than males, with an OR (0.61 (95%CI, 0.43–0.86)), *p =* 0.005 (Figure 4).

### 3.6. Homosexual vs. Heterosexual

Two studies reported the frequency of homosexual or heterosexual willingness to receive the vaccine. The pooled OR indicated that homosexuals are more willing to receive the vaccine than heterosexuals with OR (1.44 (95%CI, 1.14–1.80)), *p =* 0.002 (Figure 5).

### 3.7. COVID-19 Vaccinated vs. Unvaccinated

Three studies (n = 898) reported the COVID-19 vaccination status of patients and their willingness to receive the vaccine. The pooled OR indicated that COVID-19-vaccinated patients are more willing to be vaccinated for Mpox than unvaccinated with OR (3.57 (95% CI, 1.89–6.74)), *p =* 0.0001 (Figure 6).

### 3.8. Patients with Chronic Diseases vs. Healthy Patients

Four studies reported patients’ health status and willingness to receive the vaccine. The pooled OR indicated that patients with chronic disease are more willing to receive the vaccine than healthy ones, with an OR (1.07 (95%CI, 1.03–1.11)), *p =* 0.0009 (Figure 7).

### 3.9. Subgroup Meta-Analysis According to the Population Willingness to Receive Monkeypox Vaccine

Through a subgroup analysis according to the population of the study, we had three different populations: LGBTQ, comprising a total of 38,806 participants, with an 82.2% willingness to take the Mpox vaccine; healthcare workers, comprising a total of 7173, with a 57.5% willingness to receive the Mpox vaccine; and the general population, who comprised a total of 80,224 with a 56.67% willingness to be vaccinated against Mpox (Figure 8).

## 4. Discussion

Vaccination remains integral to public health in infectious disease prevention and control. While vaccine development and distribution present their own scientific and logistical challenges, the success of these efforts is mainly dependent on public acceptance. As seen during the COVID-19 pandemic, vaccine hesitancy, which includes delays in accepting or refusing vaccines despite their availability, poses significant threats to achieving optimal vaccination coverage [41]. In this context, the Mpox vaccine emerges as a contemporary challenge, with low intention to receive vaccines by the general public and HCWs [13,15,42], demanding urgent attention and understanding of public perceptions and acceptance levels to ensure timely prevention and educational intervention.

As reported by the WHO, it is evident that the dynamics of Mpox transmission are undergoing significant shifts. Historically endemic to West or Central Africa [43], the re-emergence of the virus in Nigeria in 2017 underscored the potential for resurgence and inter-human transmission within familiar territories. However, the sudden rise in Mpox cases in Europe by May 2022, an area previously unaffected, led to swift governmental actions, encompassing an extensive educational paradigm and an expedited vaccine dissemination strategy [44]. A previous systematic review [45] reveals differences in Mpox vaccination acceptance across various geographical regions. Over half of the 8045 participants (56.0%) indicated acceptance of the vaccine, with European countries exhibiting higher acceptance rates at 70.0% compared to Asian countries at 50.0%. In our study, the observed 56.14% Mpox vaccine acceptance rate underlines notable geographical variations, with Asia—despite its delayed Mpox incidence—dominating the research landscape at 46.67% (14/30 studies) compared to Europe (7/30, 23.33%), Africa (2/30, 6.67%), North America (4/30, 13.33%), the Middle East (2/30, 6.67%), and South America (1/30, 3.33%). Remarkably, countries like Vietnam [21] and China [17,25] showcased high acceptance rates (>90%), likely influenced by the aftermath of the COVID-19 pandemic. Conversely, persistent low acceptance rates (<30%) in nations such as Romania [23], Japan [18], Pakistan [24], and Jordan [14] highlight the urgent need for tailored public health interventions. Given these disparities, it is evident that both historical experiences with pandemics and regional contexts play pivotal roles in shaping public health attitudes [46,47], necessitating targeted vaccination strategies for each unique setting.

The analysis shows marked disparities among distinct sub-populations. Individuals identifying as homosexuals are 37% [18] and 91% [20] more willing to receive the Mpox vaccine, aligning with previous systematic review outcomes [45]. The trend is arguably anchored in the enduring societal challenges this group confronts, predominantly discrimination and diminished self-worth [48,49]. Grounded in the looking-glass self-theory [50], such experiences contribute to their more empathetic response to communal practices that benefit their community [51]. Additionally, the heightened vulnerability of LGBTQ individuals to both infectious and non-communicable diseases likely augments their adherence to vaccination guidelines [52]. This is in line with our findings that showed that the highest subgroup to be willing to be vaccinated was the LGBTQ subgroup, with an 82.2% willingness rate, compared to healthcare workers (57.5%) and the general population (56.67%). In contrast, females exhibit a reduced acceptance rate of the Mpox vaccine compared to males, lower by a margin of 15% to 60% [12,16,19,21,22,33]. Our research reveals the intricate landscape of vaccine hesitancy in women, particularly among older age groups and pregnant women [53,54,55]. Notably, pregnant women exhibit significantly higher vaccine hesitancy rates [55], influenced by potential side effects and unverified information on social media [55]. These findings suggest a greater vaccine hesitancy among women than men, which could be attributed to their intricate family roles [56,57].

Our findings also indicate a significant correlation between Mpox vaccine acceptance and prior COVID-19 vaccination, with individuals having received COVID-19 vaccines exhibiting a 2- to 5-fold increased likelihood in willingness to accept the Mpox vaccine relative to their non-vaccinated counterparts [13,15,19]. This correlation may be attributable to the pervasive influence of misinformation and conspiracy theories, which have substantially polarised vaccine decision-making from cautious endorsement to outright refusal [58,59,60]. The proliferation of unfounded information and spurious claims regarding SARS-CoV-2 vaccines on digital platforms has undermined public trust even before the authorisation of efficacious vaccines [60]. This erosion in trust has engendered a marked polarisation in vaccination perspectives, with 4% to 7% of respondents in COVID-19 surveys expressing opposition to vaccination, aligning closely with our observed Mpox vaccine rejection rate of 4.5% [61].

Other socioeconomic factors such as marital status, social income, and educational level may have some influence on the willingness to be vaccinated [62]. However, this was not clearly described in the included studies, with a huge difference in age groups, marital status subdivisions, different educational levels, and social income levels.

Furthermore, results suggest a higher willingness to receive the Mpox vaccine among individuals with underlying conditions [12,16,21,33]. People with chronic illnesses often experience heightened health-related fear and anxiety, which could lead to adverse mental effects [63,64]. This proactive seeking of disease prevention methods, driven by such concerns, could account for the higher acceptance of Mpox vaccines among individuals with chronic conditions, as they view vaccination as a constructive approach [65,66]. Additionally, recommendations for regular vaccination to mitigate complications in individuals with chronic illnesses could further promote vaccine acceptance in this group [67].

The FDA has approved two paramount vaccines for Mpox prevention: JYNNEOS (Imvamune/Imvanex) and ACAM2000 (Dryvax). The safety profiles of these vaccines are commendable, predominantly presenting minor side effects such as localised pain, redness, and swelling. Notably, in light of the persistent Mpox outbreak, the FDA, in August 2022, expanded the emergency use authorisation for JYNNEOS to include individuals aged 18 years and below [68].

Our findings underscore the imperative for national and regional bodies to intensify educational campaigns, particularly leveraging social and news media platforms pivotal in disseminating credible vaccine information [69]. The results of this systematic review and meta-analysis contribute significantly to the global measures adopted for Mpox vaccination and public health policy formulation, specifically targeting sub-populations with documented lower acceptance rates for the Mpox vaccine. Furthermore, in the face of future outbreaks, these insights could facilitate identifying and stratifying demographics resistant to vaccination, enabling tailored interventions for these particular cohorts [69,70].

We can emphasise the significance of this meta-analysis by shedding light on vaccine hesitancy, highlighting its prevalence and disparities across demographic groups. Its novelty originates from its investigation of Mpox vaccination attitudes in relation to gender, sexual orientation, and COVID-19 vaccination status, which provides new insights into these domains. However, several limitations remain. All included studies are cross-sectional studies with differences in the methodologies and tools for measurement, which might introduce heterogeneity and affect the findings’ potential to be generalised. Moreover, several confounding variables, such as population differences, exist in every study that produces heterogeneous results. Most of the included studies are single-centred ones. There are definitely many people who are vaccine ambivalent; however, this was not clearly described in the included studies as they only mentioned the willingness or refusal to receive Mpox vaccine. Therefore, this may produce some sort of bias. We recommend further multi-centre studies to assess the willingness of Mpox vaccination. Future awareness campaigns are needed to illustrate the importance of vaccination against Mpox to overcome the occurrence of another pandemic. Media coverage, political situation, or previous experiences with vaccination campaigns can have a significant impact on vaccination decisions and change the minds of those thinking that the vaccination would cause harm [70,71]. This was previously proven by Cascini et al. [72] who noted that social media and other media platforms can be used as a forum for public health interventions and as a source of data to guide policy decisions aimed at addressing vaccine hesitancy and advancing vaccination rates across the globe. It is crucial to raise vaccination rates in order to establish herd immunity among community members [73]. As studies are evolving, similar to our results (56.14%, were willing to vaccinate against mpox), a previous meta-analysis found that the figure was 56.0% [45]. Improving acceptance for monkeypox vaccination involves addressing various factors, including education, communication, community engagement, and building trust.

## 5. Conclusions

This meta-analysis has advanced our understanding of public attitudes towards Mpox vaccination by highlighting disparities and key factors influencing it. While providing useful insights, it also serves as a starting point for future research aimed at developing more effective communication strategies and ensuring equitable vaccine coverage in the light of changing socio-cultural and health contexts.

## Figures and Tables

**Figure 1 vaccines-11-01840-f001:**
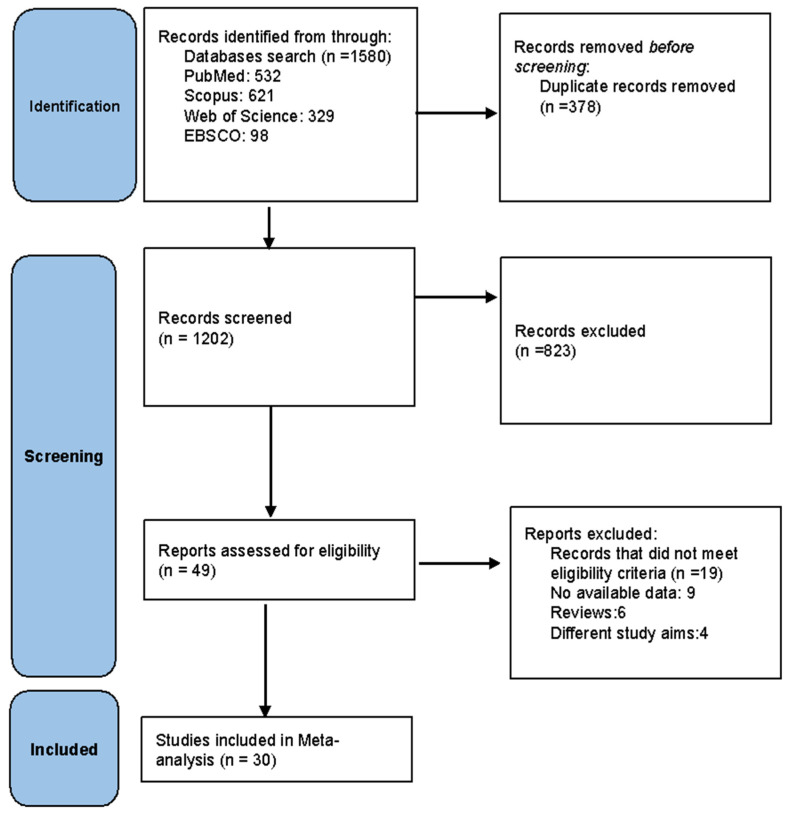
PRISMA flow diagram of the included studies.

**Figure 2 vaccines-11-01840-f002:**
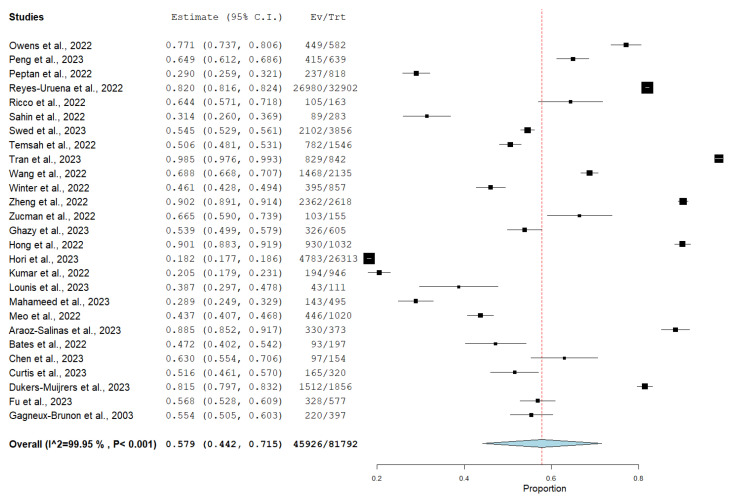
Forest plot of overall willingness among study participants to receive Mpox vaccine. (Black: each study proportion; Blue: pooled result 95%CI; Red dashed: pooled result, central value).

**Figure 3 vaccines-11-01840-f003:**
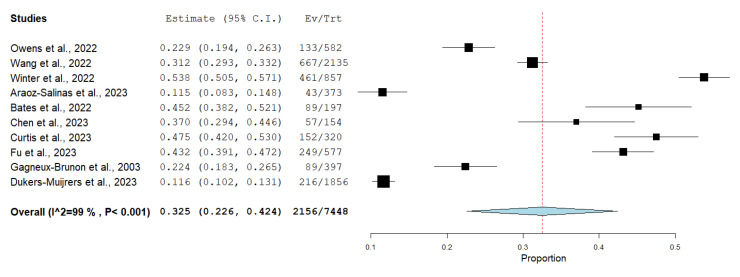
Forest plot illustrating unwillingness among study participants to receive the Mpox vaccine.(Red: central value. Blue: pooled result 95%CI; Black: central value for each study).

**Figure 4 vaccines-11-01840-f004:**
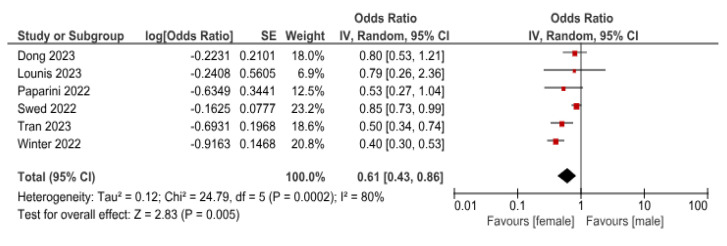
Comparative analysis of willingness to receive Mpox vaccine between female and male participants. (Red: central value; Black: pooled result 95%CI).

**Figure 5 vaccines-11-01840-f005:**
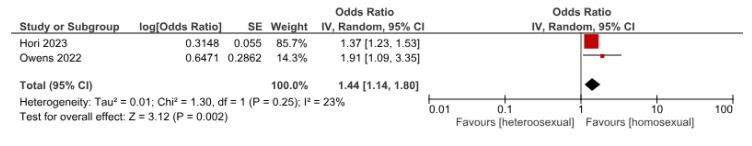
Comparison of willingness to receive Mpox vaccine between homosexual and heterosexual participants. (Red: central value; Black: pooled result 95%CI).

**Figure 6 vaccines-11-01840-f006:**
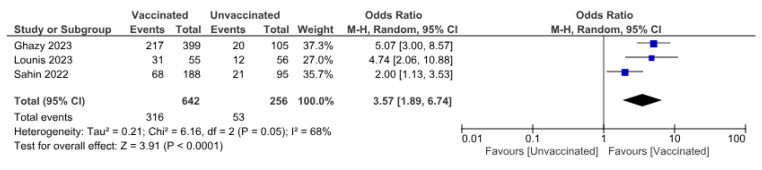
Association between COVID-19 vaccination status and willingness to receive Mpox vaccine.

**Figure 7 vaccines-11-01840-f007:**
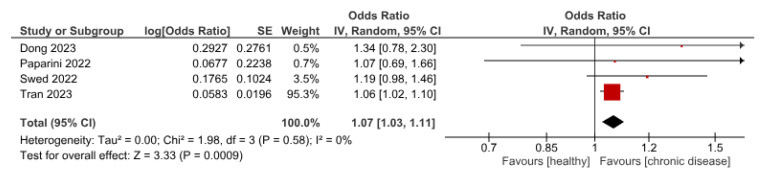
Influence of chronic diseases on participants’ willingness to receive Mpox vaccine.

**Figure 8 vaccines-11-01840-f008:**
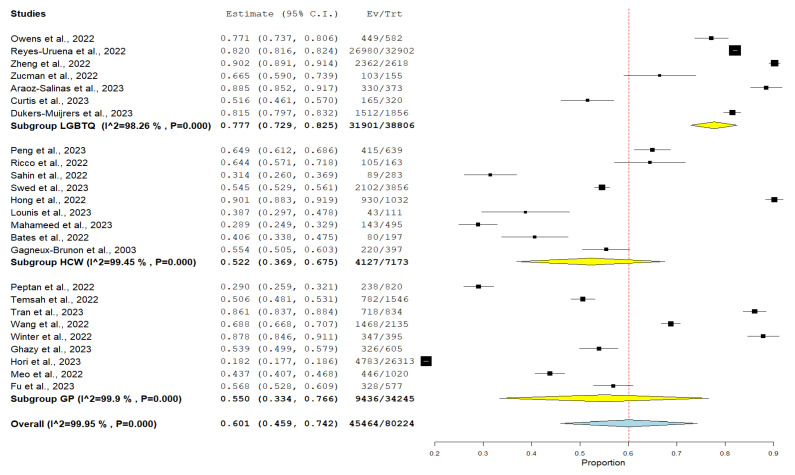
Subgroup analysis of the participants’ willingness to receive monkeypox vaccine according to their population. (Yellow: subgroup pooled análisis; Blue: general pooled result 95%CI; Red dashed line: pooled central value).

**Table 1 vaccines-11-01840-t001:** Baseline characteristics and summary of the included studies.

Study ID	Country	Study Design	Sample Size	Gender (M/F)	Age, Mean (SD)	Population	Summary of Findings
Ghazy et al., 2023 [13]	Ghana	Cross-sectional	605	368/237	30.0 ± 6.8	General population	An overall small percentage of participants were willing to receive the Mpox vaccine with a higher willingness among healthcare workers, and participants who received the COVID-19 vaccine.
Hong et al., 2022 [25]	China	Cross-sectional	1032	266/766	-	Healthcare workers	Most participants showed their intention to be vaccinated, while the minor group who refused the vaccine explained their decision as a result of their fear of the adverse effects.
Hori et al., 2023 [18]	Japan	Cross-sectional	26,313	12,900/13,413	48 ± 16.8	The general population and gays	Vaccine acceptance was higher among homosexual participants than heterosexual participants.
Kumar et al., 2022 [24]	Pakistan	Cross-sectional	946	432/514	22.5 ± 3.5	University students	Most respondents showed a neutral attitude regarding the Mpox vaccine. Awareness of Mpox was related to academic degree, study discipline, and geographic location.
Lounis et al., 2023 [19]	Algeria	Cross-sectional	111	33/78	-	Healthcare workers	One-third of participants showed a positive attitude towards Mpox vaccination, and history of COVID-19 vaccination positively correlated with willingness for Mpox vaccination.
Mahameed et al., 2023 [14]	Jordan	Cross-sectional	495	204/291	-	Healthcare workers	Most respondents showed a neutral attitude towards vaccination. But the rate of vaccine acceptance was significantly related to previous vaccine uptake.
Meo et al., 2022 [34]	Saudi Arabia	Cross-sectional	1020	466/554	-	General population	Most participants showed satisfactory knowledge about Mpox; also, there is a positive attitude towards preventive measures against Mpox.
Owens et al., 2022 [20]	United States	Cross-sectional	582	582/0	>18	MSM	The rural MSM had a lower intention to be vaccinated for Mpox than urban MSM.
Paparini et al., 2022 [33]	United Kingdom	Cross-sectional	1911	1781/112	43 ± 10	MSM	Respondents showed good knowledge about Mpox and reported high levels of understanding and acceptability of the Mpox vaccine.
Peng et al., 2023 [35]	China	Cross-sectional	639	208/431	37.91 ± 9.4	Healthcare workers	Healthcare workers in China had high awareness of Mpox and positive attitudes towards Mpox vaccination and were concerned about the Mpox epidemic.
Peptan et al., 2022 [23]	Romania	Cross-sectional	818	396/422	-	General population	High acceptance rate among participants previously vaccinated against COVID-19.
Reyes-Uruena et al., 2022 [36]	Europe	Cross-sectional	32,902	32,902/0	38 ± 9	MSM	high acceptance of Mpox vaccination among MSM who use dating apps.
Ricco et al., 2022 [1]	Italy	Cross-sectional	163	57/106	42.9 ± 10	Physicians	The majority of participants showed a favourable attitude towards the Mpox vaccine.
Sahin et al., 2022 [15]	Turkey	Cross-sectional	283	117/166	32 ± 8.8	Physicians	Less than a third of the participants planned to have the Mpox vaccine.
Swed et al., 2022 [37]	Arab countries	Cross-sectional	3665	1477/2157	-	General population	A large percentage of the respondents showed acceptance of the Mpox vaccination.
Temsah et al., 2022 [38]	KSA	Cross-sectional	1546	650/896	-	General population	Age and high education level are associated with low agreement with vaccination.
Tran et al., 2023 [21]	Vietnam	Cross-sectional	834	239/595	-	General population	Most of the participants were willing to take the vaccine, but vaccine hesitancy was mostly due to insufficient information on Mpox and the vaccine.
Wang et al., 2022 [39]	China	Cross-sectional	2135	1337/798	-	General population	Most participants were willing to take preventive measures, but higher age and income were associated with lower vaccine acceptance.
Winter et al., 2022 [22]	United States	Cross-sectional	856	410/436	-	General population	About half of the participants were willing to be vaccinated against Mpox. A strong association between previous vaccination and intention to receive a Mpox vaccine.
Zheng et al., 2022 [17]	China	Cross-sectional	2618	-	-	MSM	Most of the participants were willing to receive the vaccines.
Zucman et al., 2022 [40]	France	Cross-sectional	361	-	-	MSM	About one-third of participants showed their hesitancy to be vaccinated against Mpox and an association between the number of sexual partners and vaccine acceptance.
Araoz-Salinas et al., 2023 [26]	Peru	Cross-sectional	373	317/23/33	31 ± 9	LGBTQ community	High intention to be vaccinated against Mpox among the LGBTQ community.
Bates et al., 2022 [27]	USA	Cross-sectional	197	113/69	>18	Physicians	There’s poor knowledge, attitude, and practice among physicians towards Mpox vaccination.
Chen et al., 2023 [28]	China	Cross-sectional	154	148/6	50.9 ± 12.7	Male sex workers	There is a positive attitude towards getting vaccines among male sex workers and poor knowledge of Mpox.
Curtis et al., 2023 [29]	United States	Cross-sectional	320	257/10	90.65 ± 7.89	LGBTQ community	Socioeconomic stability, fear of getting the disease, and vaccine hesitancy were strongly associated with an individual’s Mpox vaccine taking.
Dukers-Muijrers et al., 2023 [30]	Netherlands	Cross-sectional	1856		42.6 ± 17.8	LGBTQ community	Peoples’ willingness to be vaccinated was high and they recommended low threshold options to get vaccinated, alongside clear, uniform and factual information.
Fu et al., 2023 [31]	China	Cross-sectional	577	523/54	32.7 ± 6.9	General population	More than half of the population showed a positive attitude towards getting the vaccine.
Gagneux-Brunon et al., 2003 [32]	France and Belgium	Cross-sectional	397	137/260	43.3 ± 12	Healthcare workers	There’s a low intention of receiving the Mpox vaccine among healthcare workers.
Dong et al., 2023 [12]	China	Cross-sectional	521	264/257	30.3 ± 6.7	General population	The Chinese population had relatively high knowledge of Mpox and demonstrated a willingness to receive the vaccine.
Swed et al., 2023 [16]	Arab countries	Cross-sectional	3856	1685/2171		Healthcare professionals	Most healthcare professionals have a moderate knowledge of Mpox. Furthermore, they demonstrated a low willingness to receive vaccination against Mpox.

Mpox: Monkey pox; LGBTQ: Lesbian, gay, bisexual, transgender, queer and questioning; MSM: Men who have sex with men. (M/F): Male/Female.

**Table 2 vaccines-11-01840-t002:** Quality assessment of included studies using Newcastle–Ottawa scale.

Study Name	Representativeness of the Cases (★)	Sample Size (★)	Non-Response Rate (★)	Ascertainment of the Screening/Surveillance Tool (Max★★)	The Potential Confounders Were Investigated by Subgroup Analysis or Multivariable Analysis (★)	Assessment of the Outcome (Max★★)	Statistical Test (★)	Overall Score
Araoz-Salinas et al., 2023 [26]	*	-	-	*	-	*	*	(4) moderate
Bates et al., 2022 [27]	*	*	-	-	**	**	*	(7) high
Chen et al., 2023 [28]	*	-		*	*	**	*	(6) moderate
Curtis et al., 2023 [29]	*	-	*		-	**	*	(5) moderate
Dong et al., 2023 [12]	*	*	-	*	-	*	*	(5) moderate
Dukers-Muijrers et al., 2023 [30]	*	-	-	*	-	*	*	(4) moderate
Fu et al., 2023 [31]	*				**	**	*	(6) moderate
Ghazy et al., 2023 [13]	*	*	**	*	*	*	*	(8) high
Gagneux-Brunon et al., 2003 [32]	*	-	**	-	*	*	*	(6) moderate
Hong et al., 2022 [25]	*	*	**	*	*	*	*	(8) high
Hori et al., 2023 [18]	*	-	**	*	*	*	*	(7) high
Kumar et al., 2022 [24]	*	*	**	*	*	*	-	(7) high
Lounis et al., 2023 [19]	-	-	**	-	*	*	*	(5) moderate
Mahameed et al., 2023 [14]	-	-	**	*	*	*	*	(6) moderate
Meo et al., 2022 [34]	*	-	**	*	*	*	-	(6) moderate
Swed et al., 2023 [16]	*	*	**	*	*	*	-	(7) high
Temsah et al., 2022 [38]	*	-	*	-	-	*	*	(4) moderate
Tran et al., 2023 [21]	*	*	*	-	-	*	*	(5) moderate
Wang et al., 2022 [39]	*	-	*	-	*	*	*	(5) moderate
Winter et al., 2022 [22]	*	-	*	-	**	*	*	(6) moderate
Zheng et al., 2022 [17]	*	*	*	-	-	*	*	(5) moderate
Zucman et al., 2022 [40]	*	-	*	-	-	*	*	(4) moderate
Owens et al., 2022 [20]	*	*	*	-	*	-	*	(5) moderate
Paparini et al., 2022 [33]	*	*	-	-	-	-	*	(3) low
Peng et al., 2023 [35]	*	*	-	-	-	-	*	(3) low
Peptan et al., 2022 [23]	*	*	-	-	-	-	*	(3) low
Reyes-Uruena et al., 2022 [36]	*	*	-	-	*	-	*	(4) moderate
Ricco et al., 2022 [1]	*	*	*	*	*	-	*	(6) moderate
Sahin et al., 2022 [15]	*	*	*	-	*	-	*	(5) moderate
Swed et al., 2022 [37]	-	*	*	-	*	-	*	(4) moderate

A study can be awarded a maximum of one star for each numbered item within the Selection and Outcome categories. A maximum of two stars can be given for Comparability.(Depending on the category, a study can be awarded 1 star/asterisk, or 2. On the botton that info is placed: A study can be awarded a maximum of one star for each numbered item within the Selection and Outcome categories. A maximum of two stars can be given for Comparability).

## Data Availability

The data that support the findings of this study are available from the corresponding author upon reasonable request.

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
