# Peer review of "Attitudes towards Receiving Monkeypox Vaccination: A Systematic Review and Meta-Analysis"

_vaccines, 2023, doi:10.3390/vaccines11121840_

Round 1

Reviewer 1 Report

Comments and Suggestions for Authors

The paper under review presents a systematic review and meta-analysis of observational studies exploring public attitudes towards Mpox vaccination. The study's primary objective is to understand the factors contributing to vaccine acceptance or hesitancy, which is crucial for the success of immunization programs.

Drawbacks:

  1. The paper highlights differences in willingness based on gender and sexual orientation. However, it does not delve into other potential factors, such as age, socioeconomic status, education level, and cultural or religious beliefs, which could significantly influence vaccination attitudes.
  2. The study provides an odds ratio for willingness between females and males and homosexuals and heterosexuals. Still, it does not offer a comparative analysis between other demographic groups, which might have provided a more nuanced understanding.
  3. The paper does not discuss the potential influence of external factors, such as media coverage, political climate, or previous experiences with vaccination campaigns, on public attitudes.

Recommendations:

  1. It would be beneficial to explore the influence of other demographic factors on vaccination attitudes, such as age, socioeconomic status, and cultural beliefs, to develop more targeted communication strategies.
  2. It would be beneficial to explore the role of media, political climate, and historical vaccination campaigns in shaping public attitudes towards Mpox vaccination.
  3. It is recommended to expand the conclusions section, including a contribution of the research, its novelty, limitations, and future research perspectives.

The paper provides valuable insights into public attitudes towards Mpox vaccination, highlighting the importance of understanding and addressing vaccine hesitancy. While the study has its limitations, it offers a foundation for future research and the development of effective health communication strategies. Addressing the identified drawbacks and implementing the recommendations can significantly enhance the impact and relevance of the study in the broader context of public health.

Author Response

Reviewer’s comment:

It would be beneficial to explore the influence of other demographic factors on vaccination attitudes, such as age, socioeconomic status, and cultural beliefs, to develop more targeted communication strategies.

Author’s reply:

Thanks for your comment. We addressed this part by subgroup analysis of the populations, and the rest in the discussion section.

Reviewer’s comment:

It would be beneficial to explore the role of media, political climate, and historical vaccination campaigns in shaping public attitudes towards Mpox vaccination.

Author’s reply:

Thanks for your comment. We do believe their role in vaccination. We also discussed this part in the discussion and recommendations.

Reviewer’s comment:

It is recommended to expand the conclusions section, including a contribution of the research, its novelty, limitations, and future research perspectives.

Author’s reply:

Thanks for your comment. We added this part.

Reviewer 2 Report

Comments and Suggestions for Authors

Thank you for the opportunity to review this work. This is an interesting manuscript. The authors conducted a systematic review and meta-analysis of observational studies evaluating public’s willingness to receive vaccination for Mpox disease. Introduction section is structured well and rationale of the study has been sufficiently described. Authors have adequately described the methods and results are also appropriately discussed in the manuscript. All in all, the manuscript is well-written for the most part, however, there are few points that needs addressed. Therefore, I recommend minor revisions for this manuscript.

Figure 1: Better to mention number of articles for each database.

Figure 1: Remove box labelled “records not retrieved” as it doesn’t contain any useful information.

Figure 1: Provide reasons for excluding 19 studies during the screening procedure.

Line 147-179; better to describe regions than the individual name of countries

Line 150; replace “university studies” with “university students”

Please include a separate column for NOS cumulative score in the Table 3.

In figure 1, authors stated that thirty studies were included in the meta-analysis. However, at line 158, authors are referring to 27 studies only. Please indicate clearly how many studies were included in the systematic review and how may were included in the meta-analysis. Provide reasons for excluding studies from meta-analysis as well.

Line 158-161, “56.1% were willing to take the vaccine”. Around 44% were vaccine rejecters? How many were the fence-sitters (vaccine ambivalent)?

Line 159-161: It is a bit confusing. Were there only 10 studies that reported rate of vaccine refusal among study participants? Authors are requested to revise this section to make it comprehensible for the readers.

Line 181-184: Please mention odds ratio and their 95% CI and p-value in parenthesis before period symbol.

Consider conducting sub-group analysis for healthcare professionals and other populations.

Author Response

Reviewer’s comment:

Figure 1: Better to mention number of articles for each database.

Author’s reply:

Thanks for your comment. We mentioned this.

Reviewer’s comment:

Figure 1: Remove box labelled “records not retrieved” as it doesn’t contain any useful information.

Author’s reply:

Thanks for your comment. We removed this box.

Reviewer’s comment:

Figure 1: Provide reasons for excluding 19 studies during the screening procedure.

Author’s reply:

Thanks for your comment. We added the reasons for exclusion.

Reviewer’s comment:

Line 147-179; better to describe regions than the individual name of countries.

Author’s reply:

Thanks for your comment. We added the regions.

Reviewer’s comment:

Line 150; replace “university studies” with “university students”

Author’s reply:

Thanks for your comment. We corrected it.

Reviewer’s comment:

Please include a separate column for NOS cumulative score in the Table 3.

Author’s reply:

Thanks for your comment. We added the column.

Reviewer’s comment:

In figure 1, authors stated that thirty studies were included in the meta-analysis. However, at line 158, authors are referring to 27 studies only. Please indicate clearly how many studies were included in the systematic review and how may were included in the meta-analysis. Provide reasons for excluding studies from meta-analysis as well.

Author’s reply:

Thanks for your comment. We corrected this part.

Reviewer’s comment:

Line 158-161, “56.1% were willing to take the vaccine”. Around 44% were vaccine rejecters? How many were the fence-sitters (vaccine ambivalent)?

Author’s reply:

Thanks for your comment. We do believe that many people are fence-sitters but this wasn’t clearly explained in the included studies so we couldn’t include it in the meta-analysis. We clarified this part in the limitations section.

Reviewer’s comment:

Line 159-161: It is a bit confusing. Were there only 10 studies that reported rate of vaccine refusal among study participants? Authors are requested to revise this section to make it comprehensible for the readers.

Author’s reply:

Thanks for your comment. We clarified this part.

Reviewer’s comment: Line 181-184: Please mention odds ratio and their 95% CI and p-value in parenthesis before period symbol.

Author’s reply: Thanks for your comment. We corrected this.

Reviewer’s comment: Consider conducting sub-group analysis for healthcare professionals and other populations.

Author’s reply: Thanks for your comment. We added this sub-group analysis.